# Exploring Asymmetric Encoder-Decoder Structure for Context-based Sentence Representation Learning

## Abstract

Context information plays an important role in human language understanding, and it is also useful for machines to learn vector representations of language. In this paper, we explore an asymmetric encoder-decoder structure for unsupervised context-based sentence representation learning. As a result, we build an encoder-decoder architecture with an RNN encoder and a CNN decoder, and we show that neither an autoregressive decoder nor an RNN decoder is required. We further combine a suite of effective designs to significantly improve model efficiency while also achieving better performance. Our model is trained on two different large unlabeled corpora, and in both cases transferability is evaluated on a set of downstream language understanding tasks. We empirically show that our model is simple and fast while producing rich sentence representations that excel in downstream tasks.

## 1 Introduction

Learning distributed representations of sentences is an important and hard topic in both the deep learning and natural language processing communities, since it requires machines to encode a sentence with rich language content into a fixed-dimension vector filled with continuous values. We are interested in learning to build a distributed sentence encoder in an unsupervised fashion by exploiting the structure and relationship in a large unlabeled corpus. Since humans interpret sentences by composing from the meanings of the words, we decompose the task of learning a sentence encoder into two essential components: learning distributed word representations, and learning how to compose a sentence representation from the representations of words in the given sentence.

Numerous studies in human language processing have claimed that the context in which words and sentences are understood plays an important role in human language understanding (Altmann & Mirkovic, 2009; Binder & Desai, 2011). The idea of learning from the context information (Turney & Pantel, 2010) was recently successfully applied to vector representation learning for words in Mikolov et al. (2013); Pennington et al. (2014).

Collobert et al. (2011) proposed a unified framework for learning language representation from the unlabeled data, and it is able to generalize to various NLP tasks. Inspired by the prior work on incorporating context information into representation learning, Kiros et al. (2015) proposed the Skip-thought model, which is an encoder-decoder model for unsupervised sentence representation learning. The paper exploits the semantic similarity within a tuple of adjacent sentences as supervision, and successfully built a generic, distributed sentence encoder. Rather than applying the conventional autoencoder model, the skip-thought model tries to reconstruct the surrounding 2 sentences instead of the input sentence. The learned sentence representation encoder outperforms previous unsupervised pretrained models on the evaluation tasks with no finetuning, and the results are comparable to the models which were trained directly on the datasets in a supervised fashion.

The usage of 2 independent decoders in Skip-thought model matches our intuition that, given the current sentence, inferring the previous sentence and inferring the next one should be different. Recently, Tang et al. (2017) proposed the Skip-thought Neighbor model, which only decodes the next

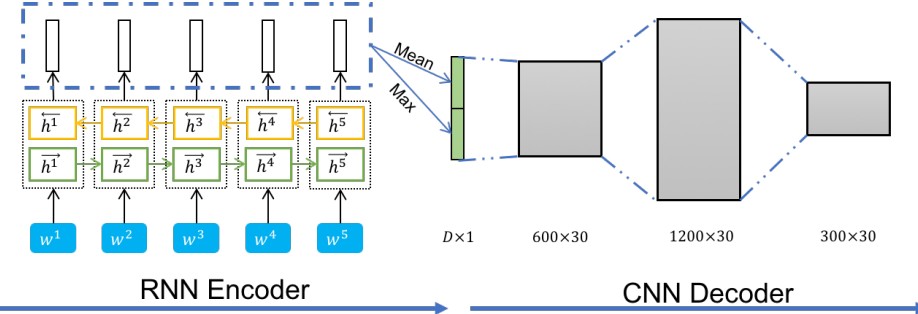

Figure 1: Our proposed model is composed of an RNN encoder, and a CNN decoder. During training, a batch of sentences are sent to the model, and the RNN encoder computes a vector representation for each of sentences; then the CNN decoder needs to reconstruct the paired target sequence, which contains 30 contiguous words right after the input sentence, given the vector representation. 300 is the dimension of word vectors. $D$ is the dimension of sentence representation, and it varies along with the change of the RNN encoder size. (Better view in color.)

sentence, and the performance on the downstream tasks is similar to that of their implementation of the Skip-thought model.

In this paper, we follow the idea in the Skip-thought Neighbor model, which exploits the subsequent context information for learning representation, and aim to bring asymmetry into structure design as well. Our proposed model has an asymmetric encoder-decoder structure, which keeps an RNN as the encoder and has a CNN as the decoder, and will be referred to as an "RNN-CNN" model. The key components of our model design can be summarized as:

1. a bidirectional RNN encodes the input sentence, and a CNN decodes all words in the paired target sequence at once, which speeds up the training process;

2. the supervision for training comes from inferring the next contiguous words given the current sentence, which helps the model to learn from the context in an unsupervised fashion;

3. the mean+max pooling captures complex interactions among words, which augments the transferability of the proposed model;

4. tying word embeddings in the encoder with the word prediction layer in the decoder constrains the input and output space to be the same, which also reduces number of parameters and regularizes the model.

We demonstrate the transferability of our model by evaluation on various downstream tasks, and the performance shows that our model improves both results and training efficiency.

## 2 RNN-CNN Model

Our model is highly asymmetric in terms of both training pairs and model structure. Specifically, our model has an RNN as the encoder, and a CNN as the decoder. During training, the encoder takes the $i$-th sentence $s_i$ as input, and then generates a fixed-dimension vector $\mathbf{z}_i$ as the sentence representation; the decoder is applied to reconstruct the next sentence or the subsequent few contiguous words $t_i$. The difference of the generated sequence and the target sequence is measured by cross-entropy loss. An illustration is in Figure 1. (For simplicity, we omit the subscript $i$ in the section.)

**Encoder:** The encoder is a bi-directional Gated Recurrent Unit (GRU) (Chung et al., 2014). We experimented with both Long-short Term Memory (LSTM, Hochreiter & Schmidhuber (1997)) and GRU. Since LSTM didn't give us significant performance boost, and generally GRU runs faster than LSTM, in our experiments, we stick to using GRU in the encoder. Suppose that a sentence $s$ contains $M$ words, which are $w^1, w^2, ..., w^M$, and they are transformed by an embedding matrix $\mathbf{E}$ to word vectors. The bi-directional GRU will take one word vector at a time, and run in both forward and backward direction; both sets of hidden states are concatenated to form the hidden state matrix

$\mathbf{H} = [\mathbf{h}^1, \mathbf{h}^2, ..., \mathbf{h}^M] \in \mathbb{R}^{D \times M}$, where $d$ is the dimension of the representations $\mathbf{h}^m = \left[ \overleftarrow{\mathbf{h}^m}; \overrightarrow{\mathbf{h}^m} \right]$ ($\forall m \in \{1, 2, ..., M\}$).

**Representation:** We aim to provide a model with faster training speed with better transferability than existing algorithms, thus we choose to apply a parameter-free composition function, which is a concatenation of the outputs from a global mean pooling over time and a global max pooling over time, on the computed sequence of hidden states. The composition function can be represented as

$$\mathbf{z} = \left[ \frac{1}{M} \sum_{m=1}^{M} \mathbf{h}^m; \max \mathbf{H}_{1.}; \max \mathbf{H}_{2.}; ...; \max \mathbf{H}_{d.} \right], \tag{1}$$

where $\max \mathbf{H}_{d.}$ is the max operation on the $d$-th row of the matrix $\mathbf{H}$, which outputs a scalar. Thus the representation $\mathbf{z}$ has a dimension of $2d$.

**Decoder:** The decoder is a 3-layer CNN to reconstruct the paired target sequence $t$, which needs to expand $\mathbf{z}$ from length 1 to the length of $t$. Intuitively, the decoder could be a stack of deconvolution layers. For fast training speed, we optimized the architecture to make it plausible to use fully-connected layers and convolution layers in the decoder, since generally, convolution layers run faster than deconvolution layers in modern deep learning frameworks.

Suppose that the target sequence $t$ has $N$ words, the first layer of deconvolution will expand $\mathbf{z}$, which could be considered as a sequence with length 1, into a feature map with length $N$. It can be easily implemented as a concatenation of outputs from $N$ linear transformations in parallel. Then the second and third layer are 1D-convolution layers with kernel size 3 and 1, respectively. The output feature map $\mathbf{V} = [\mathbf{v}^1, \mathbf{v}^2, ..., \mathbf{v}^N]$, where $\mathbf{v} \in \mathbb{R}^e$, and $e$ is dimension of the word vectors.

Note that our decoder is not an autoregressive model, and it brings us high training efficiency. We will discuss the reason of choosing this decoder which we call a predict-all-words CNN decoder.

**Objective:** A softmax layer is applied after the decoder to produce a probability distribution over words at each position, $\mathbf{softmax}(\mathbf{E}\mathbf{v}^n)$, and the training objective is to minimize the sum of the negative log-likelihood over all positions in the target sequence $t$:

$$\mathcal{L} = -\sum_{n=1}^{N} \log P(w^n | \mathbf{z}). \tag{2}$$

The loss function $\mathcal{L}$ is summed over all sentences in the training corpus.

## 3 ARCHITECTURE DESIGN

We follow the idea of an encoder-decoder model with using the context information for learning sentence representations in an unsupervised fashion. Since the decoder won't be used after training, and the quality of the generated sequences is not our main focus, it is important to study the design of the decoder. Generally, a fast training algorithm is preferred, thus proposing a new decoder with high training efficiency and also strong transferability is crucial for an encoder-decoder model.

### 3.1 CNN AS THE DECODER

Our design of the decoder is basically a 3-layer ConvNet, and it predicts all words in the next sequence all at once. In contrast, existing work, such as Skip-thought Kiros et al. (2015), and CNN-LSTM Gan et al. (2017),use autoregressive RNNs as the decoders.

An autoregressive model is good at generating sequences with high quality, such as language and speech. However, an autoregressive decoder seems to be unnecessary in an encoder-decoder model for learning sentence representations, since it won't be used after training, and it runs quite slow during training. Therefore, we conducted experiments to test the necessity of using an autoregressive decoder in learning sentence representations, and we had 2 findings.

**Finding I: It is not necessary to input the correct words into an autoregressive decoder in terms of learning good sentence representations.**

| Decoder | SICK-r | SICK-E | STS14 | MSRP (Acc/F1) | SST | TREC |
|---|---|---|---|---|---|---|
| **auto-regressive RNN as decoder** | | | | | | |
| Baseline | 0.8530 | 82.6 | 0.51/0.50 | 74.1 / 81.7 | 82.5 | 88.2 |
| Always Sampling | 0.8576 | 83.2 | 0.55/0.53 | 74.7 / 81.3 | 80.6 | 87.0 |
| Uniform Sampling | 0.8559 | 82.9 | 0.54/0.53 | 74.0 / 81.8 | 81.0 | 87.4 |
| **auto-regressive CNN as decoder** | | | | | | |
| Baseline | 0.8510 | 82.8 | 0.49/0.48 | 74.7 / 82.8 | 81.4 | 82.6 |
| Always Sampling | 0.8535 | 83.3 | 0.53/0.52 | 75.0 / 81.7 | 81.4 | 87.6 |
| Uniform Sampling | 0.8568 | 83.4 | 0.56/0.54 | 74.7 / 81.4 | 83.0 | 88.4 |
| **predict-all-words RNN as decoder** | | | | | | |
| RNN | 0.8508 | 82.8 | 0.58/0.55 | 74.2 / 82.8 | 81.6 | 88.8 |
| **predict-all-words CNN as decoder** | | | | | | |
| CNN | 0.8530 | 82.6 | 0.58/0.56 | **75.6** / 82.9 | 82.8 | 89.2 |
| CNN-Max | 0.8465 | 82.6 | 0.50/0.47 | 73.3 / 81.5 | 79.1 | 82.2 |
| Double-sized RNN Encoder | | | | | | |
| CNN | **0.8631** | **83.9** | **0.58**/0.55 | 74.7 / **83.1** | **83.4** | **90.2** |
| CNN-Max | 0.8485 | 83.2 | 0.47/0.44 | 72.9 / 80.8 | 82.2 | 86.6 |

Table 1: The models here all have a bi-directional GRU as the encoder (dimensionality 300 in each direction). The default way of producing the representation is a concatenation of outputs from a global mean-pooling and a global max-pooling, while "·-*Max*" refers to the model with only global max-pooling. Bold numbers are the best results among all presented models. We found that **1)** inputting correct words to an autoregressive decoder is not necessary; **2)** predict-all-words decoders work roughly the same as autoregressive decoders; **3)** mean+max pooling provides stronger transferability than the max-pooling alone does. The table supports our choice of the predict-all-words CNN decoder and the way of producing vector representations from the bi-directional RNN encoder.

The experimental design was inspired by Bengio et al. (2015). The model we designed for the experiment has a bi-directional GRU as the encoder, and an autoregressive decoder, including both RNN and CNN. We started by analyzing the effect of different sampling strategies of the input words on learning an auto-regressive decoder.

We compared 3 autoregressive decoding settings: 1) using ground-truth words (**Baseline**), 2) using previously predicted words (**Always Sampling**), and 3) using uniformly sampled words from the dictionary (**Uniform Sampling**). The 3 decoding settings were named by Bengio et al. (2015). The results are presented in the Table 1.

Generally, the three different decoding settings didn't make much of a difference in terms of the performance on selected downstream tasks, with RNN or CNN as the decoder. The results tell us that, in terms of learning good sentence representations, the autoregressive decoder doesn't require the correct ground-truth words as the inputs.

**Finding II: The model with an autoregressive decoder works roughly the same as the model with a predict-all-words decoder.**

With Finding I, we noticed that the correct ground-truth input words to the autoregressive decoder is not necessary in terms of learning sentence representations. Therefore, it makes sense to test whether we need an autoregressive model at all.

In our model, the CNN decoder predicts all words at once during training, which is different from autoregressive decoders, and we call it a predict-all-words CNN decoder. We want to compare the performance of the predict-all-words decoders and that of the autoregressive decoders separate from the RNN/CNN distinction, thus we designed a predict-all-words CNN decoder and RNN decoder.

The predict-all-words CNN decoder is described in Section 2, which is a stack of 3 convolutional layers, and all words are predicted once at the output of the decoder. The predict-all-words RNN decoder is built based on our CNN decoder. To keep the number of parameters roughly the same, we replaced the last 2 convolutional layers with a bidirectional GRU.

The results are also presented in the Table 1. The performance of the predict-all-words RNN decoder does not significantly differ from that of any one of the autoregressive RNN decoders, and the same observation was observed in CNN decoders.

These two findings actually support our choice of using a predict-all-words CNN as the decoder, and it brings the model higher training efficiency and strong transferability.

## 3.2 Mean+Max Pooling

Since the encoder is a bi-directional RNN in our model, we have multiple ways to select/compute on the generated hidden states to produce a sentence representation. In Skip-thought (Kiros et al., 2015) and SDAE (Hill et al., 2016), only the hidden state at the last time step produced by the RNN encoder is regarded as the vector representation for a given sentence, which may not be the most expressive vector for representing the input sentence.

We followed the idea proposed in Chen et al. (2016). They built a model for supervised SNLI task (Bowman et al., 2015) that concatenates the outputs from a global mean pooling and a global max pooling to serve as a sentence representation, and showed a performance boost on the SNLI dataset. Also, Conneau et al. (2017) found that the model with global max pooling function has stronger transferability than the model with a global mean pooling function after supervised training on SNLI.

In our proposed RNN-CNN model, we empirically show that the mean+max pooling provides stronger transferability than the max pooling does, and the results are presented in Table 1. The concatenation of a mean-pooling and a max pooling function is actually a parameter-free composition function, and the computation load is negligible compared to heavy matrix multiplications. Also, the non-linearity of the max pooling function augments the mean pooling function for building a representation that captures a more complex composition of the syntactic information.

## 3.3 Tying Word Embeddings and Word Prediction Layer

We choose to share the parameters in the word embedding layer in RNN encoder and the word prediction layer in CNN decoder. The tying was proposed in both Press & Wolf (2017) and Inan et al. (2016), and it generally helps to learn a better language model. In our model, the tying also drastically reduces the number of parameters, which could prevent overfitting.

Furthermore, we initialize the word embeddings with pretrained word vectors, such as word2vec (Mikolov et al., 2013) and GloVe (Pennington et al., 2014), since it has been shown that these pretrained word vectors can serve as good initialization for deep learning models, and more likely lead to better results than random samples from a uniform distribution.

## 3.4 Study of the Hyperparameters in Our Model Design

We studied hyperparameters in our model design based on 3 out of 10 downstream tasks, including SICK-r, SICK-E (Marelli et al., 2014), and STS14 (Agirre et al., 2014). The first model we created, which is reported in Section 2, is a decent design, and the following variations didn't give us much performance change except small improvements with increasing the dimensionality of the encoder. However, we think it is worth mentioning the effect of hyperparameters in our model design. We present the Table in the supplementary material and we summarize it as follows:

1. Decoding the next sentence worked similarly as decoding the subsequent contiguous words.

2. Decoding subsequent 30 words, which was adopted from the Skip-thought training code [1], gave us a reasonable good performance. More words for decoding didn't give us a significant performance gain, while it took longer to train.

3. Adding more layers into the decoder and enlarging the dimension of the convolutional layers indeed sightly improved the performance on the 3 downstream tasks, but as training efficiency is one of our main concerns, we decided it wasn't worth sacrificing training efficiency for the minor performance improvement.

---

[1] https://github.com/ryankiros/skip-thoughts/blob/master/training/train.py

4. Increasing the dimensionality of the RNN encoder improved the model performance, and the additional training time brought by it was less than that by adding more layers and enlarging the dimension of the convolutional layers in the CNN decoder. We reported results from both smallest and largest models in Table 2.

## 4    Experiment Settings

The large corpus we used for unsupervised training is the BookCorpus dataset Zhu et al. (2015), which contains 74 million sentences from 7000 books in total. For stable training, we use ADAM (Kingma & Ba, 2014) algorithm for optimization, and gradient clipping (Pascanu et al., 2013) when the norm of gradient exceeds a certain value. Since we didn't find any significant difference between word2vec and GloVe as initialization in terms of the performance, we stick to using the word vectors from word2vec to initialize the word embedding layer in our models.

The vocabulary for unsupervised training contains the top 20k most frequent words in BookCorpus. In order to generalize the model trained with a relatively small, fixed vocabulary to the much larger set of all possible English words, Kiros et al. (2015) proposed a word expansion method that learns a linear projection from the pretrained word embeddings word2vec to the learned RNN word embeddings. Thus, the model benefits from the generalization ability of the pretrained word embeddings.

The downstream tasks for evaluation include semantic relatedness (SICK) (Marelli et al., 2014), paraphrase detection (MSRP) (Dolan et al., 2004), question-type classification (TREC) (Li & Roth, 2002), and 5 benchmark sentiment and subjective datasets, which includes movie review sentiment (MR, SST) (Pang & Lee, 2005; Socher et al., 2013), customer product reviews (CR) (Hu & Liu, 2004), subjectivity/objectivity classification (SUBJ) (Pang & Lee, 2004), opinion polarity (MPQA) (Wiebe et al., 2005), and semantic textual similarity (STS14) (Agirre et al., 2014). After unsupervised training on the BookCorpus dataset, we fix the parameters in the encoder, and apply it as a sentence representation extractor on the 10 tasks.

In order to compare the effect of different corpora, we also trained 2 models on Amazon Book Review dataset (without ratings) which is the largest subset of the Amazon Review dataset (McAuley et al., 2015) with 142 million sentences after tokenization, about twice as large as BookCorpus.

Both training and evaluation of our models were conducted in PyTorch [2], and we used SentEval [3] provided by Conneau et al. (2017) to evaluate the transferability of models with different settings. All the models were trained for the same number of iterations with the same batch size, and the performance was measured at the end of training for each of the models.

## 5    Related work and Comparison

Table 2 presented the results on 10 evaluation tasks of our proposed RNN-CNN models, and related work. "**small RNN-CNN**" refers to the model with the dimension of representation as 1200, and "**large RNN-CNN**" refers to that as 4800. The results of our model on SNLI can be found in Table 3.

Our work was inspired by analyzing the **Skip-thought** model (Kiros et al., 2015). Skip-thought model successfully applied this form of learning from the context information into unsupervised representation learning for sentences, in which the model learns to encode the current sentence and decode the surrounding 2 sentences, and then, Ba et al. (2016) augmented the LSTM with proposed layer-normalization (**Skip-thought+LN**), which improved the skip-thought model generally on all downstream tasks. Instead of applying RNNs in the model, Hill et al. (2016) proposed the **FastSent** model which only learns source and target word embeddings, and it is a generalization of CBOW (Mikolov et al., 2013) to sentence-level learning, and the composition function over word embeddings is a summation operation. Later on, Gan et al. (2017) applied a CNN as the encoder, which is called the **CNN-LSTM** model. The proposed composition model follows the idea of encoding the current sentence and predicting itself and the next sentence; the proposed hierarchical model leverages the context information from both sentence-level and paragraph-level, while learning to encode the current sentence and predict the next one, the model has another RNN to process the sentence representation one at a time at paragraph-level.

---

[2]http://pytorch.org/
[3]https://github.com/facebookresearch/SentEval

| Model | Hrs | SICK-$r$ | SICK-E | STS14 | MSRP | TREC | MR | CR | SUBJ | MPQA | SST |
|---|---|---|---|---|---|---|---|---|---|---|---|
| *Unsupervised training with unordered sentences* | | | | | | | | | | | |
| Unigram-TFIDF | - | - | - | - | **73.6/81.7** | **85.0** | 73.7 | 79.2 | 90.3 | 82.4 | - |
| ParagraphVec | 4 | - | - | 0.42/0.43 | 72.9/81.1 | 59.4 | 60.2 | 66.9 | 76.3 | 70.7 | - |
| word2vec BOW | 2 | **0.8030** | **78.7** | 0.65/0.64 | 72.5/81.4 | 83.6 | 77.7 | **79.8** | 90.9 | **88.3** | 79.7 |
| fastText BOW | - | 0.8000 | 77.9 | 0.63/0.62 | 72.4/81.2 | 81.8 | 76.5 | 78.9 | **91.6** | 87.4 | 78.8 |
| GloVe BOW | - | 0.8000 | 78.6 | 0.54/0.56 | 72.1/80.9 | 83.6 | **78.7** | 78.5 | **91.6** | 87.6 | **79.8** |
| SDAE | 72 | - | - | 0.37/0.38 | 73.7/80.7 | 78.4 | 74.6 | 78.0 | 90.8 | 86.9 | - |
| *Unsupervised training with ordered sentences - BookCorpus* | | | | | | | | | | | |
| DiscSent‡ | 8 | - | - | - | 75.0/ - | 87.2 | - | - | 93.0 | - | - |
| FastSent | 2 | - | - | **0.63/0.64** | 72.2/80.3 | 76.8 | 70.8 | 78.4 | 88.7 | 80.6 | - |
| FastSent+AE | 2 | - | - | 0.62/0.62 | 71.2/79.1 | 80.4 | 71.8 | 76.5 | 88.8 | 81.5 | - |
| Skip-thought | 336 | 0.8580 | 82.3 | 0.29/0.35 | 73.0/82.0 | 92.2 | 76.5 | 80.1 | 93.6 | 87.1 | 82.0 |
| Skip-thought+LN | 720 | 0.8580 | 79.5 | 0.44/0.45 | - | 88.4 | 79.4 | **83.1** | 93.7 | 89.3 | 82.9 |
| combine CNN-LSTM | - | 0.8618 | - | - | **76.5/83.8** | **92.6** | 77.8 | 82.1 | 93.6 | 89.4 | - |
| *small RNN-CNN*† | 20 | 0.8530 | 82.6 | 0.58/0.56 | 75.6/82.9 | 89.2 | 77.6 | 80.3 | 92.3 | 87.8 | 82.8 |
| *large RNN-CNN*† | 34 | 0.8698 | 85.2 | 0.59/0.57 | 75.1/83.2 | 92.2 | **79.7** | 81.9 | **94.0** | 88.7 | **84.1** |
| *Unsupervised training with ordered sentences - Amazon Book Review* | | | | | | | | | | | |
| *small RNN-CNN*† | 21 | 0.8476 | 82.7 | **0.53/0.53** | 73.8/81.5 | 84.8 | 83.3 | 83.0 | 94.7 | 88.2 | 87.8 |
| *large RNN-CNN*† | 33 | 0.8616 | 84.3 | 0.51/0.51 | **75.7/82.8** | 90.8 | 85.3 | 86.8 | 95.3 | 89.0 | 88.3 |
| *Unsupervised training with ordered sentences - Amazon Review* | | | | | | | | | | | |
| BYTE m-LSTM | 720 | 0.7920 | - | - | 75.0/**82.8** | - | **86.9** | 91.4 | 94.6 | 88.5 | - |
| *Supervised training - Transfer learning* | | | | | | | | | | | |
| NMT En-to-Fr | 72 | - | - | 0.43/0.42 | - | 82.8 | 64.7 | 70.1 | 84.9 | 81.5 | - |
| CaptionRep BOW | 24 | - | - | 0.46/0.42 | - | 72.2 | 61.9 | 69.3 | 77.4 | 70.8 | - |
| DictRep BOW | 24 | - | - | 0.67/**0.70** | 68.4/76.8 | 81.0 | 76.7 | 78.7 | 90.7 | 87.2 | - |
| BiLSTM-Max(SNLI) | <24 | **0.8850** | 84.6 | 0.68/0.65 | 75.1/82.3 | **88.7** | 79.9 | 84.6 | 92.1 | 89.8 | 83.3 |
| BiLSTM-Max(AllNLI) | <24 | 0.8840 | **86.3** | **0.70/0.67** | **76.2/83.1** | 88.2 | **81.1** | **86.3** | **92.4** | **90.2** | **84.6** |
| *Supervised task-dependent training - No transfer learning* | | | | | | | | | | | |
| NB-SVM | - | - | - | - | - | - | 79.4 | 81.8 | 93.2 | 86.3 | 83.1 |
| AdaSent | - | - | - | - | - | 92.4 | 83.1 | 86.3 | 95.5 | 93.3 | - |
| Tree-LSTM | - | 0.8680 | - | - | - | - | - | - | - | - | - |
| TF-KLD | - | - | - | - | 80.4/85.9 | - | - | - | - | - | - |

Table 2: **Related Work and Comparison.** As presented in the table, our designed asymmetric RNN-CNN model has strong transferability, and is overall better than existing unsupervised models in terms of fast training speed and good performance on evaluation tasks. The table presents the model comparison. "†"s refer to our models, and "*small*/*large*" refers to the dimension of representation as 1200/4800. "‡" indicates that DiscSent model was trained with additional data from Wikipedia and the Gutenberg project. Bold numbers are the best ones among the models with same training and transferring setting, and underlined numbers are best results among all unsupervised representation learning models. For STS14, the performance measures are Pearson's and Spearman's score. For MSRP, the performance measures are accuracy and F1 score.

Our model falls in the same category as it is an encoder-decoder model. However, we aim to propose an efficient and effective model. Instead of decoding the surrounding 2 sentences as in Skip-thought, FastSent and the compositional CNN-LSTM, our model only decodes the subsequent sequence with a fixed length. Compared with hierarchical CNN-LSTM, our model showed that, with a proper model design, this next-words context information is sufficient in learning sentence representations. Particularly, our proposed **small RNN-CNN** model runs roughly 3 times faster than our implemented Skip-thought model on the same GPU machine during training.

Another unsupervised approach is to learn a discriminative model by distinguishing whether a target sentence is in the context of the source sentence, and also the discourse information. **DiscSent** (Jernite et al., 2017) proposed to learn a classifier on top of the representations, which judges 1) whether the two sentences are adjacent to each other, 2) whether the two sentences are in the correct order, and 3) whether the second sentence starts with a conjunction phrase. **DisSent** (Nie et al., 2017) pointed out that human annotated explicit discourse relations is also good for learning sentence representations. It is a very promising research direction since the proposed models are generally

computational efficient and have clear intuition. However, the performance on the downstream tasks is still worse than encoder-decoder models.

Proposed by Radford et al. (2017), **BYTE m-LSTM** model uses a multiplicative LSTM unit (Krause et al., 2016) to learn a language model on Amazon Review data McAuley et al. (2015). The model works reasonably well on the downstream tasks, since the RNNs are able to produce a distributed representation for the given left-context information, such as a sentence or a document. In our experiment, we also trained our RNN-CNN model on the Amazon Book review, which is the largest subset of the Amazon review dataset, and indeed, we had a performance gain on all single-sentence classification tasks. The performance gain in our experiment and also in BYTE m-LSTM was brought by the matching between the corpus domain and the domain of downstream tasks, and it raises 2 questions 1) which corpus is good for learning sentence representations, and 2) whether the downstream tasks are comprehensive to cover sufficient aspects of a sentence.

Previously mentioned models are learned from ordered sentences, but unordered sentences can also be used for learning representations of sentences. **ParagraphVec** (Le & Mikolov, 2014) learns a fixed-dimension vector for each sentence by predicting the words within the given sentence. However, after training, the representation for a new sentence is hard to derive, since it requires optimizing the sentence representation towards an objective. **SDAE** (Hill et al., 2016) learns the sentence representations with a denoising auto-encoder model. The noise was added in the encoder by replacing words with a fixed token, and swapping two words, both with a specific probability. Our proposed RNN-CNN model trains faster than SDAE does, since the CNN decoder runs faster than the RNN decoder in SDAE, and since we utilized the sentence-level continuity as a supervision which SDAE doesn't, our model largely performs better than SDAE.

| Model | SNLI (Acc %) |
|---|---|
| *Unsupervised Transfer Learning* | |
| Skip-thought (Vendrov et al.) | 81.5 |
| large RNN-CNN *BookCorpus* | **81.7** |
| large RNN-CNN *Amazon* | 81.5 |
| *Supervised Training* | |
| ESIM (Chen et al.) | 86.7 |
| DIIN (Gong et al.) | **88.9** |

Table 3: We implemented the same classifier as mentioned in Vendrov et al. (2015) on top of the features computed by our model. Our proposed RNN-CNN model gets similar result on SNLI as Skip-thought, but with much less training time.

Supervised transfer learning is also promising when we are able to get large enough labeled data. Conneau et al. (2017) applied a bi-directional LSTM as the sentence encoder with multiple fully-connected layers to deal with both SNLI (Bowman et al., 2015), and MultiNLI (Williams et al., 2017). The trained model demonstrates a very impressive transferability on all downstream tasks, including both supervised and unsupervised. The direct and discriminative training signal pushes the RNN encoder to focus on the semantics of a given sentence, which learns to a boost in performance, and beats all other methods. Our RNN-CNN model trained on Amazon Book Review data has better results on supervised classification tasks than **BiLSTM-Max** does, while the performance of ours on semantic relatedness tasks is inferior to BiLSTM-Max. We argue that labeling a large amount of training data is time-consuming and costly; unsupervised learning could potentially provide a great initial point for human labeling making it less costly and more efficient.

## 6 CONCLUSION

Inspired by learning to exploit the contextual information present in adjacent sentences, we proposed an asymmetric encoder-decoder model with a suite of techniques for improving context-based unsupervised sentence representation learning. Since we believe that a simple model will be faster in training and easier to analyze, we opt to use simple techniques in our proposed model, including 1) an RNN as the encoder, and a predict-all-words CNN as the decoder, 2) learning by inferring next contiguous words, 3) mean+max pooling, and 4) tying word vectors with word prediction. With thorough discussion and extensive evaluation, we justify our decision making for each component in our RNN-CNN model. In terms of the performance and the efficiency of training, we justify that our model is a fast and simple algorithm for learning generic sentence representations from unlabeled corpora. Further research will focus on how to maximize the utility of the context information, and how to design simple architectures to best make use of it.

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

# Supplementary

**Anonymous authors**

| Encoder | | Decoder | | Hrs | SICK-$r$ | SICK-E | STS14 | MSRP (Acc/F1) | SST | TREC |
|---|---|---|---|---|---|---|---|---|---|---|
| type | dim | type | dim | | | | | | | |
| **Dimension of Sentence Representation: 1200** | | | | | | | | | | |
| RNN 2x300 | | CNN | 600-1200-300 | 20 | 0.8530 | 82.6 | 0.58/0.56 | **75.6/82.9** | 82.8 | **89.2** |
| | | CNN$^\dagger$ | 600-1200-300 | 21 | 0.8515 | 82.7 | 0.58/0.56 | 75.3/82.5 | **82.9** | 85.2 |
| | | CNN(10) | 600-1200-300 | 11 | 0.8474 | 82.9 | 0.57/0.55 | 74.2/81.6 | 82.8 | 88.0 |
| | | CNN(50) | 600-1200-300 | 27 | 0.8533 | 82.5 | 0.57/0.55 | 74.7/82.2 | 81.5 | 86.2 |
| RNN 2x300 | | RNN | 600 | 26 | 0.8530 | 82.6 | 0.51/0.50 | 74.1/81.7 | 81.0 | 89.0 |
| CNN 4x300$^\S$ | | CNN | 600-1200-300 | 8 | 0.8117 | 80.5 | 0.44/0.42 | 72.7/80.7 | 78.4 | 85.0 |
| RNN 2x300 | | CNN | 600-1200-2400-300 | 28 | **0.8570** | **84.0** | 0.58/0.56 | 74.3/81.5 | 82.8 | 88.2 |
| | | CNN | 1200-2400-300 | 27 | 0.8541 | 83.0 | **0.59/0.57** | 74.3/82.2 | **82.9** | 89.0 |
| **Dimension of Sentence Representation: 2400** | | | | | | | | | | |
| RNN 2x600 | | CNN | 600-1200-300 | 25 | 0.8631 | 83.9 | **0.58/0.55** | **74.7/83.1** | 83.4 | **90.2** |
| RNN 2x600 | | RNN | 600 | 32 | **0.8647** | **84.2** | 0.52/0.51 | 74.0/81.2 | **84.2** | 87.6 |
| CNN 3x800$^\ddagger$ | | RNN | 600 | 8 | 0.8132 | - | - | 71.9/81.9 | - | 86.6 |
| **Dimension of Sentence Representation: 4800** | | | | | | | | | | |
| RNN 2x1200 | | CNN | 600-1200-300 | 34 | **0.8698** | **85.2** | **0.59/0.57** | **75.1/83.2** | 84.1 | **92.2** |
| Skip-thought (Kiros et al., 2015) | | | | 336 | 0.8584 | 82.3 | 0.29/0.35 | 73.0/82.0 | 82.0 | **92.2** |
| Skip-thought+LN (Ba et al., 2016) | | | | 720 | 0.8580 | 79.5 | 0.44/0.45 | - | 82.9 | 88.4 |

Table 1: **Architecture Comparison**. As shown in the table, our designed asymmetric RNN-CNN model (row 1,9, and 12) works better than other asymmetric models (CNN-LSTM, row 11), and models with symmetric structure (RNN-RNN, row 5 and 10). In addition, with larger encoder size, our model demonstrates stronger transferability. The default setting for our CNN decoder is that it learns to reconstruct 30 words right next to every input sentence. "CNN(10)" represents a CNN decoder with the length of outputs as 10, and "CNN(50)" represents it with the length of outputs as 50. "$\dagger$" indicates that the CNN decoder learns to reconstruct next sentence. "$\ddagger$" indicates the results reported in Gan et al. as *future predictor*. The CNN encoder in our experiment, noted as "$\S$", was based on AdaSent in Zhao et al. and Conneau et al.. Bold numbers are best results among models at same dimension, and underlined numbers are best results among all models. For STS14, the performance measures are Pearson's and Spearman's score. For MSRP, the performance measures are accuracy and F1 score.

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
