# OpenReview forum: "Exploring Asymmetric Encoder-Decoder Structure for Context-based Sentence Representation Learning"
_ICLR.cc/2018/Conference — Reject_

### Official Review · AnonReviewer3 · 2017-11-19
**Good results, but tons of minor issues**

**Rating:** 6
**Confidence:** 4

**Review:**

Update:

I'm going to change my review to a 6 to acknowledge the substantial improvements you've made—I no longer fear that there are major errors in the paper, but this paper is still solidly borderline, and I'm not completely convinced that any new claim is true. The evidence presented for the main claim—that you can get by without an autoregressive decoder when pretraining encoders—is somewhat persuasive, but isn't as unequivocal as I'd hope, and even if the claim is true, it is arguably too limited a claim for an ICLR main conference paper. As R1 says, a *ACL short paper would be more appropriate.  The writing is also still unclear in places.

----

This paper presents a new RNN encoder–CNN decoder hybrid design for use in pretraining reusable sentence encoders on Kiros's SkipThought objective. The task is interesting and important, and the results are generally good: The new model outperforms SkipThought, and all other prior models for training sentence encoders on unlabeled data. However, some of the design choices seem a bit odd, and I have a large number of minor concerns about the paper. I'd like to see the authors' replies and the other reviews before I can confidently endorse this paper as correct.


Non-autoregressive decoding with a CNN strikes me as a somewhat ill-posed problem, even for in this case where you don't actually use the decoder in the final application of your model. At each position, you're training your model to predict a distribution over all words that could appear at the beginning/tenth position/twentieth position in sentences on some topic. I'd appreciate some more discussion of why this should or shouldn't hurt performance. I'd be less concerned about this if the results supporting the use of the CNN decoder were a bit more conclusive: while they are better on average across your smaller experiments, your largest experiment (2400D) shows them roughly tied.

Your paper opens with the line "Context information plays an important role in human language understanding." This sounds like it's making an empirical claim that your paper doesn't support, but it's so vague that it's hard to tell exactly what that claim is. Please clarify this or remove it.

This sentence is quite inaccurate: "The idea of learning from the context information was first successfully applied to vector representation learning for words in Mikolov et al. (2013b) and learning from the occurrence of words also succeeded in Pennington et al. (2014)." Turney and Pantel 2010 ( https://www.jair.org/media/2934/live-2934-4846-jair.pdf ) offer a survey of the substantial prior work that existed at that time.

The "Neighborhood Hypothesis" is given quite a lot of importance, given that it's a fairly small empirical effect without any corresponding theory. The fact that it's emphasized so heavily makes me suspect that I can guess the author of the paper. I'd tone down that part of your framing.

I would appreciate some more analysis of which of the non-central tricks that you describe in section 3 help. For example, max pooling seems reasonable, but you report yourself that mean pooling generally works much better in prior work. Without an explicit experiment, it's not clear why you'd add a mean pooling component.

It seems misleading to claim that your CNN is modeled after AdaSent, as that model uses a number of layers that varies with the length of the sentence (and differs from yours in a few other less-important ways). Please correct or clarify.

The use of “†” in table to denote models that predict the next sentence in a sequence doesn't make sense. It should apply to all of your models if I understand correctly. Please clarify.

You could do a better job at table placement and formatting. Table 3 is in the wrong section, for example.

You write that: "Our proposed RNN-CNN model gets similar result on SNLI as Skip-thought, but with much less training time." This seems to be based on a comparison between your model run on your hardware and their model run on their (presumably older) hardware, and possibly also with their older version of CuDNN. If that's right, you should tone down this claim or offer some more evidence.

---

> ### Author Response · Authors · 2017-12-19
> **Thanks a lot for your review and your useful questions.**
>
> (I) We chose to focus on the encoder-decoder model for learning vector representations of sentences in an unsupervised fashion, and after training, only the encoder will be used to produce a vector for a given input sentence. Since the decoder won’t be applied after training, and faster training time of the models is generally preferred, it makes sense to use a non-autoregressive decoder. Thus, we proposed to use a predict-all-words CNN decoder.
>
> As suggested by Reviewer 2 at the top of this page, we conducted experiments to support our choice of the decoder and had 2 findings.
>
> 1/   In terms of learning good sentence representations, for an autoregressive model, RNN or CNN, as the decoder in an encoder-decoder model, it is not necessary to input the ground-truth words to the decoder during training.
> 2/ The model with an autoregressive decoder works roughly the same as the model with a predict-all-words decoder in learning sentence representations
>
> The 2 findings from our experiments show that using a predict-all-word CNN as the decoder works as well as an autoregressive RNN as the decoder. In addition, a predict-all-word CNN decoder runs fast during training.
>
>
> (II) Mean+Max Pooling vs Max Pooling
>
> We didn’t report that the mean pooling works better than the max pooling in our paper, and we also agree that the max pooling is reasonable.
>
> In our paper, our claim is that a combination of max pooling and mean pooling works better than max pooling, which was inspired by Chen et al., 2016[1]. In order to consolidate the claim, we conducted experiments to compare “max pooling” and “mean+max pooling”. The results are presented in the Table below:
>
>
> Encoder (TrainHrs)            | SICK-r   SICK-E     STS14      | MSRP (ACC/F1) |    SST      TREC
>
>                                                        600D Representation
> RNN-max (21hrs)              | 0.8365    82.6       0.50/0.47  |      73.3 / 81.5     |   79.1        82.2
>
>                                                        1200D Representation
> RNN-max  (28 hrs)            | 0.8485    83.2       0.47/0.44   |      72.9 / 80.8     |  82.2        86.6
> RNN-mean+max (21 hrs) | 0.8530    82.6       0.58/0.56   |      75.6 / 82.9     |  82.8       89.2
>
>
> As we can see, the model trained with mean+max pooling generally works better than it with max pooling only. We also evaluated the model, which is trained with only max pooling over time, with mean+max pooling during testing (with no additional weight training), and it boosts the performance on the unsupervised evaluation task, and also gets slightly better results on all supervised evaluation tasks, which also supports our claim that mean+max pooling works better than max pooling.
>
>
> (III) Clarification of the CNN decoder
>
> Our CNN decoder is not modeled after AdaSent, and it is a stack of 3 convolutional layers.
>
> In section 3, we tried to compare whether to use an RNN encoder or a CNN encoder, and the CNN ENCODER  we designed here is based on the design in Conneau et al., (2017) [2], since it is a modification of AdaSent, and it performs better than other RNN models except for the RNN-Max model on SNLI dataset. By consulting other papers, we confirmed that the CNN encoder we designed for comparison is also a good model.
>
> “†” is used to indicate the model that predicts the next sentence, and all of our other models are learned to predict next 30 words, not necessarily a sentence. In Section 3, we compared the performance of our model that predicts the next sentence, and that of our model that predicts the next contiguous 30 words. The results showed that there is no significant difference between the 2 models.
>
>
> (V) Our implementation of Skip-thought on the same GPU machine
>
> We reimplemented the skip-thought model in PyTorch, and ran it on the same GPU machine used for this paper. The training of our implemented Skip-thought model took approximately 3 times longer than is needed for our model.
>
> Our understanding of the reason why the Skip-thought model is slow is that 1) it has 2 decoders to decode the previous sentence and the next one respectively, 2) the 2 autoregressive RNN decoders run slowly during training, 3) the size of the RNN decoders is fairly large.
>
> We addressed these 3 issues by proposing a predict-all-words CNN decoder, which is a non-autoregressive decoder.
>
> [1]Chen, Qian et al. “Enhancing and Combining Sequential and Tree LSTM for Natural Language Inference.” CoRR abs/1609.06038 (2016): n. pag.
> [2] Conneau, Alexis et al. “Supervised Learning of Universal Sentence Representations from Natural Language Inference Data.” EMNLP (2017).

---

### Official Review · AnonReviewer1 · 2017-11-23
**The contribution of the paper is to extend Skip-thought by 1) decoding only one target sentence; 2) using a CNN decoder.**

**Rating:** 3
**Confidence:** 5

**Review:**

The authors build on the work of Tang et al. (2017), who made a minor change to the skip-thought model by decoding only the next sentence, rather than the previous one also. The additional minor change in this paper is to use a CNN, rather than RNN, decoder.

I am sympathetic to the goals of the work, and believe this sort of work should be carried out, but I see the contribution as too minor to constitute a paper at the conference track of a leading international conference such as ICLR. Given the incremental nature of the work, I think this would be a good fit for something like a short paper at *ACL.

I found the more theoretical motivation of the CNN decoder not terribly convincing, and somewhat post-hoc. I feel as though analogous arguments could just as easily be made for an RNN decoder. Ultimately I see these questions - such as CNN vs. RNN for the decoder - as empirical ones.

Finally, the authors have admirably attempted a thorough comparison with existing work, in the related work section, but this section takes up a large chunk of the paper at the end, and again I would have preferred this section to be much shorter and more concise.

Summary: worthwhile empirical goal, but the paper could have been easily written using half as much space.

---

> ### Author Response · Authors · 2017-12-19
> **Thanks for your critical review, and we really appreciate your thoughts about our paper.**
>
> We agree that carrying out research in learning sentence representations with the encoder-decoder model is important, and proposing new models is even more exciting, but we still wanted to argue that, analyzing previously proposed models and building efficient learning algorithms based on previous findings are also important.
>
> Our paper differs from existing works in the following important ways:
>
> 1/ We aim to propose an efficient encoder-decoder model for learning sentence representations in an unsupervised fashion, and the prior work didn’t pay much attention to the running time. In our paper, the running time is also a consideration for model selection.
>
> 2/ The proposed model in our paper is not a simple modification from the Skip-thought model or the model proposed in Tang et al., (2017). We have a focus on simplifying the decoder.
>
> 3/ Our paper, with new experiments suggested by Reviewers 2 and 3, has a unique axis for comparison. Our findings suggest that for learning sentence representations with an encoder-decoder model, it is not necessary to use an autoregressive model for decoding.
>
> 4/ In addition, the 3x speedup can greatly increase the sizes of models that can be run, the number
> of experiments that can be done,  and also makes the model accessible to those without the best computational resources.  The recent popularity of deep learning and LSTM demonstrates the power that making an algorithm easier and faster to run can have.
>
> The overall goal of our paper is to propose an efficient encoder-decoder model for learning sentence representation in an unsupervised fashion. Since only the encoder will be used after training, it makes sense to simplify the decoder to make it run faster, and help the model perform even better.
>
> The key difference between the previously proposed encoder-decoder models and our model lies in the choice of decoder, and as you pointed out, most of the previous models adopted an autoregressive RNN model for decoding, while we proposed to use a predict-all-words CNN for decoding. The autoregressive models, including RNNs and CNNs, are good at generating sequential data, such as language and voice, but in our case, the quality of the generated sequences after training is not our main focus, since we care about the quality of the learned sentence representations. Thus, it is not necessary to use an autoregressive model for decoding.
>
> Based on this idea (and now backed by new experiments suggested by Reviewers  2  and 3), we proposed to use a predict-all-words CNN decoder instead of an autoregressive RNN decoder.
>
> (The experimental design is described in our reply to Reviewer 2, and also will be included in our updated paper.)
>
> Briefly, the results show that, for learning sentence representations with an encoder-decoder model,
>
> 1) if we stick to using an autoregressive decoder, including RNNs and CNNs, it is not necessary to input the ground-truth words to the decoder during training, and the performance on downstream tasks stays roughly the same for RNN decoder, and gets slightly better for CNN decoder;
>
> 2) the model with an autoregressive decoder performs similarly with that with a predict-all-words decoder.
>
> These 2 findings actually support our choice of using a predict-all-words CNN as the decoder, and it brings the model higher training efficiency and strong transferability.
>
> In our paper, we also develop tricks which boost the performance on the downstream tasks, such as mean+max pooling, and weight tying between word embedding and prediction layer.
>
> We agree that our comparison should be more comprehensive and reasonable, and the writing should be more concise. We will update our paper very soon.
>
> Overall, we think that our paper has its own unique theoretical considerations and empirical contributions with the suggestions from all 3 reviewers, and it should be solid and comprehensive to merit a publication at ICLR conference.

---

### Official Review · AnonReviewer2 · 2017-11-29
**Compelling results with a (relatively) efficient encoder-decoder architecture for sentence embedding learning**

**Rating:** 7
**Confidence:** 4

**Review:**


-- updates to review: --

Thanks for trying to respond to my comments. I find the new results very interesting and fill in some empirical gaps that I think were worth investigating. I'm now more confident that this paper is worth publishing and I increased my rating from 6 to 7.

I admit that this is a pretty NLP-specific paper, but to the extent that ICLR has core NLP papers (I think it does have some), I think the paper is a reasonable fit for ICLR. It might feel more at home at a *ACL conference though.

-- original review is below: --

This paper is about modifications to the skip-thought framework for learning sentence embeddings. The results show performance comparable to or better than skip-thought while decreasing training time.

I think the overall approach makes sense: use an RNN encoder because we know it works well, but improve training efficiency by changing the decoder to a combination of feed-forward and convolutional layers.

I think it may be the case that this works well because the decoder is not auto-regressive but merely predicts each word independently. This is possible because the decoder will not be used after training. So all the words can be predicted all at once with a fixed maximum sentence length. In typical encoder-decoder applications, the decoder is used at test time to get predictions, so it is natural to make it auto-regressive. But in this case, the decoder is thrown away after training, so it makes more sense to make the decoder non-auto-regressive.  I think this point should be made in the paper.

Also, I think it's worth noting that an RNN decoder could be used in a non-auto-regressive architecture as well. That is, the sentence encoding could be mapped to a sequence of length 30 as is done with the CNN decoder currently; then a (multi-layer) BiLSTM could be run over that sequence, and then a softmax classifier can be attached to each hidden vector to predict the word at that position. It would be interesting to compare that BiLSTM decoder with the proposed CNN decoder and also to compare it to a skip-thought-style auto-regressive RNN decoder. This would let us understand whether the benefit is coming more from the non-auto-regressive nature of the decoder or from the CNN vs RNN differences.

That is, it would make sense to factor the decision of decoder design along multiple axes. One axis could be auto-regressive vs predict-all-words. Another axis could be using a CNN over the sequence of word positions or an RNN over the sequence of word positions.  For auto-regressive models, another axis could be train using previous ground-truth word vs train using previous predicted word.  Skip-thought corresponds to an auto-regressive RNN (using the previous ground-truth word IIRC).  The proposed decoder is a predict-all-words CNN.  It would be natural to also experiment with an auto-regressive CNN and a predict-all-words RNN (like what I described in the paragraph above). The paper is choosing a single point in the space and referring to it as a "CNN decoder" whereas there are many possible architectures that can be described this way and I think it would strengthen the paper to increase the precision in discussing the architecture and possible alternatives.

Overall, I think the architectural choices and results are strong enough to merit publication. Adding any of the above empirical comparisons would further strengthen the paper.

However, I did have quibbles with some of the exposition and some of the claims made throughout the paper. They are detailed below:

Sec. 2:

In the "Decoder" paragraph: please add more details about how the words are predicted. Are there final softmax layers that provide distributions over output words? I couldn't find this detail in the paper. What loss is minimized during training? Is it the sum of log losses over all words being predicted?

Sec. 3:

Section 3 does not add much to the paper. The motivations there are mostly suggestive rather than evidence-based. Section 3 could be condensed by about 80% or so without losing much information. Overall, the paper has more than 10 pages of content, and the use of 2 extra pages beyond the desired submission length of 8 should be better justified. I would recommend adding a few more details to Section 2 and removing most of Section 3. I'll mention below some problematic passages in Section 3 that should be removed.

Sec. 3.2:
"...this same constraint (if using RNN as the decoder) could be an inappropriate constraint in the decoding process."  What is the justification or evidence for this claim?  I think the claim should be supported by an argument or some evidence or else should be removed. If the authors intend the subsequent paragraphs to justify the claim, then see my next comments.

Sec. 3.2:
"The existence of the ground-truth current word embedding potentially decreases the tendency for the decoder to exploit other information from the sentence representation."
But this is not necessarily an inherent limitation of RNN decoders since it could be addressed by using the embedding of the previously-predicted word rather than the ground-truth word. This is a standard technique in sequence-to-sequence learning; cf. scheduled sampling (Bengio et al., 2015).

Sec. 3.2:
"Although the word order information is implicitly encoded in the CNN decoder, it is not emphasized as it is explicitly in the RNN decoder. The CNN decoder cares about the quality of generated sequences globally instead of the quality of the next generated word. Relaxing the emphasis on the next word, may help the CNN decoder model to explore the contribution of context in a larger space."
Again, I don't see any evidence or justification for these arguments. Also see my discussion above about decoder variations; these are not properties of RNNs vs CNNs but rather properties of auto-regressive vs predict-all-words decoders.

Sec. 5.2-5.3:
There are a few high-level decisions being tuned on the test sets for some of the tasks, e.g., the length of target sequences in Section 5.2 and the number of layers and channel size in Section 5.3.

Sec. 5.4:
When trying to explain why an RNN encoder works better than a CNN encoder, the paper includes the following: "We stated above that, in our belief, explicit usage of the word order information will augment the transferability of the encoder, and constrain the search space of the parameters in the encoder. The results match our belief."
I don't think these beliefs are concrete enough to be upheld or contradicted. Both encoders explicitly use word order information. Can you provide some formal or theoretical statement about how the two encoders treat word order differently? I fear that it's only impressions and suppositions that lead to this difference, rather than necessarily something formal.

Sec. 5.4:
In Table 1, it is unclear why the "future predictor" model is the one selected to be reported from Gan et al (2017). Gan et al has many settings and the "future predictor" setting is the worst. An explanation is needed for this choice.

Sec. 6.1:

In the "BYTE m-LSTM" paragraph:

"Our large RNN-CNN model trained on Amazon Book Review (the largest subset of Amazon Review) performs on par with BYTE m-LSTM model, and ours works better than theirs on semantic relatedness and entailment tasks."  I'm not sure this "on par" assessment is warranted by the results in Table 2.  BYTE m-LSTM is better on MR by 1.6 points and better on CR by 4.7 points. The authors' method is better on SUBJ by 0.7 and better on MPQA by 0.5.  So on sentiment tasks, BYTE m-LSTM is clearly better, and on the other tasks the RNN-CNN is typically better, especially on SICK-r.


More minor things are below:

Sec. 1:
The paper contains this: "The idea of learning from the context information was first successfully applied to vector representation learning for words in Mikolov et al. (2013b)"

I don't think this is accurate. When restricting attention to neural network methods, it would be more correct to give credit to Collobert et al. (2011). But moving beyond neural methods, there were decades of previous work in using context information (counts of context words) to produce vector representations of words.

typo: "which d reduces" --> "which reduces"

Sec. 2:
The notation in the text doesn't match that in Figure 1: w_i^1 vs. w_1 and h_i^1 vs h_1.

Instead of writing "non-parametric composition function", describe it as "parameter-free". "Non-parametric" means that the number of parameters grows with the data, not that there are no parameters.

In the "Representation" paragraph: how do you compute a max over vectors? Is it a separate max for each dimension? This is not clear from the notation used.

Sec. 3.1:
inappropriate word choice: the use of "great" in "a great and efficient encoding model"

Sec. 3.2:
inappropriate word choice: the use of "unveiled" in "is still to be unveiled"

Sec. 3.4:
Tying input and output embeddings can be justified with a single sentence and the relevant citations (which are present here). There is no need for speculation about what may be going on, e.g.: "the model learns to explore the non-linear compositionality of the input words and the uncertain contribution of the target words in the same space".

Sec. 4:
I think STS14 should be defined and cited where the other tasks are described.

Sec. 5.3:
typo in Figure 2 caption: "and and"


Sec. 6.1:

In the "Skip-thought" paragraph:

inappropriate word choice: "kindly"

The description that says "we cut off a branch for decoding" is not clear to me. What is a "branch for decoding" in this context? Please modify it to make it more clear.


References:

Bengio S, Vinyals O, Jaitly N, Shazeer N. Scheduled sampling for sequence prediction with recurrent neural networks. NIPS 2015.

Collobert R, Weston J, Bottou L, Karlen M, Kavukcuoglu K, Kuksa P. Natural language processing (almost) from scratch. Journal of Machine Learning Research 2011.

---

> ### Author Response · Authors · 2017-12-19
> **Thanks for your comprehensive review and your very helpful constructive comments.**
>
> (I) Autoregressive models vs. Predict-all-words models
>
> Based on your suggestion, we conducted experiments to empirically justify our choice of the decoder. In our experiments, we have 2 findings in terms of learning sentence representations:
>
> 1)  In an encoder-decoder model with an autoregressive RNN or CNN as the decoder, it is not necessary to input the correct words to the decoder.
> 2) The model with an autoregressive decoder works roughly the same as the model with a predict-all-words decoder.
>
> The experimental design is described in detail below:
>
> 1} We compared 3 autoregressive decoding settings: 1) using ground-truth words (Baseline), 2) using previously predicted words (Always Sampling), and 3) using uniformly sampled words from the dictionary (Uniform Sampling). The 3 decoding settings were inspired by Bengio et al. 2015[1]. The results are presented in the table below:
>
> Generally, 3 decoding settings didn’t make much of a difference in terms of the performance on downstream tasks, with RNN  OR  CNN as the decoder. The results tell us that, in terms of learning good sentence representations, the autoregressive decoder doesn’t require the ground-truth words as the inputs.
>
>
> Model                       | SICK-r   SICK-E     STS14       | MSRP (ACC/F1) |    SST      TREC
>
>                                   auto-regressive RNN as  decoder
>
> Baseline                  | 0.8530   82.6       0.51/0.50   |      74.1 / 81.7       |   82.5        88.2
> Always Sampling   | 0.8576   83.2       0.55/0.53   |      74.7 / 81.3       |   80.6        87.0
> Uniform Sampling| 0.8559   82.9       0.54/0.53   |      74.0 / 81.8       |   81.0        87.4
>
>                                    auto-regressive CNN as  decoder
>
> Baseline                   | 0.8510   82.8       0.49/0.48   |      74.7 / 82.8       |  81.4         82.6
> Always Sampling    | 0.8535   83.3       0.53/0.52   |      75.0 / 81.7       |  81.4         87.6
> Uniform Sampling | 0.8568   83.4       0.56/0.54   |      74.7 / 81.4       |  83.0        88.4
>
>                                    predict-all-words RNN as decoder
>
> RNN                          | 0.8508   82.8       0.58/0.55   |      74.2 / 82.8        |  81.6         88.8
>
>                                    predict-all-words CNN as decoder
>
> CNN                          | 0.8530   82.6       0.58/0.56   |      75.6 / 82.9        |  82.8         89.2
>
>
> 2} The predict-all-words CNN decoder is described in our paper, which is a stack of 3 convolutional layers, and all words are predicted once at the output of the decoder. The predict-all-words RNN decoder is built based on our CNN decoder. To keep the number of params roughly the same, we replaced the last 2 conv layers with a bidirectional GRU.
>
> The results are also presented in the table above. The performance of the predict-all-words RNN/CNN decoder does not significantly differ from that of any one of the autoregressive RNN/CNN decoders.
>
> We aim to propose a new model with high training efficiency and strong transferability, thus a predict-all-words CNN decoder is our top choice.
>
>
> (II) Clarifications
>
> A1) The softmax layer is used to provide a word distribution at every position, and a sum of log losses is calculated for all words in the next sentence.
>
> A2) To avoid overfitting, we picked 3 tasks  (SICK-r, SICK-E and STS14) instead of all tasks to tune the high-level decisions of our model. However, the first model (row 1 in Table 1) we built works the best, and the following hyperparameter tuning (row 2,3,4,7 and 8) doesn’t boost perform.
>
> In row 7, we added another conv layer in the decoder, and it gave us slightly better performance. However, the training efficiency is also a concern, so it is not worth sacrificing the training efficiency for a slight performance gain.
>
> A3) In our design, the model only encodes the current sentence and then decodes the next one, which is stated as the “future predictor” in Gan et al. (2017)[2]. In their design, the encoder is a CNN instead of an RNN.
>
> They found that the “future predictor” worked badly overall, so they need to incorporate higher-level information to augment the “future predictor” model. In our model, we use an RNN for encoding, and it works much better than their “future predictor”.
>
> In Table 2, we did include the best results in Gan et al. (2017)[2] to compare with our models (row 12 labeled “combine CNN-LSTM”, which is an ensemble of their proposed 2 models)
>
> A4) The Skip-thought model uses 2 decoders to decode the previous sentence and the next one. Compared with the skip-thought model, we only applied one decoder to decode the next sentence, and got better results than the skip-thought model.
>
> (III) Additional comments will be addressed in the updated paper.
>
> [1] Bengio, Samy et al. “Scheduled Sampling for Sequence Prediction with Recurrent Neural Networks.” NIPS (2015).
> [2] Gan, Zhe et al. “Learning Generic Sentence Representations Using Convolutional Neural Networks.” EMNLP (2017).

---

> > ### Comment · AnonReviewer2 · 2018-01-03
> > **thanks**
> >
> > Thanks for the additional experimental results and the clarifications!

---

### Public Comment · ~Samuel_R._Bowman1 · 2017-11-01
**Data volume question**

Just out of curiosity, do you have any results on how the quantity of unlabeled training data you use impacts model performance?

---

> ### Author Response · Authors · 2017-11-01
> **Thanks and we are working it.**
>
> Thank you for your reply. It is a great question. Currently, we don’t have the results for your question, but it’ll be good to see the effect of the quantity of unlabeled training data.
>
> For clarification, we trained models on BookCorpus (74 million) and Amazon Book Review (142 million), respectively, and they were both trained with the same number of iterations and batch size. The results indicate that the model trained on Amazon Book Review outperforms that on BookCorpus. We think that the performance boost was mostly brought by the domain matching between the evaluation tasks and the training data, not by the amount of training data in different corpora.
>
> We’ll start our experiments on the effect of the quantity of unlabeled training data and get back with results very soon.

---

> > ### Public Comment · ~Samuel_R._Bowman1 · 2017-11-02
> > **...**
> >
> > Thanks! It's certainly reasonable to just report one number (no other similar paper reports learning curves that I know of), but a learning curve would help to at least suggest an answer to an interesting question: Could you get even better results with another order of magnitude more data/training time?

---

> > > ### Author Response · Authors · 2017-11-03
> > > **Results**
> > >
> > > We trained our small RNN-CNN model with 4 different amounts of unlabeled data from BookCorpus, which are 20%, 10%, 5%, and 2% of total data, respectively. All the models were evaluated on the SICK-r, and SICK-Entailment (supervised) and STS14 (unsupervised). The results are presented in the table below:
> > >
> > > Percentage      sick-r   sick-E(%)    sts14(Pearson/Spearman)
> > > 2%                    0.8347    81.4               0.59/0.57
> > > 5%                    0.8367    81.1               0.60/0.58
> > > 10%                  0.8415    81.7               0.60/0.58
> > > 20%                  0.8528    82.1               0.59/0.56
> > > 100%                0.8530    82.6              0.58/0.56
> > >
> > > We are not able to copy the performance curve of each model during training to this discussion forum, but we will update this curve into our paper in the future revised version. Here, we report some interesting observations.
> > >
> > > 1/ Longer training time helps all the models learn better representations for sentences, but the performance of each model converges after a certain number of iterations, which matches the Figure 2 in our paper. Therefore, there is no need to train the model for an unlimited time.
> > >
> > > Unexpectedly, the models trained with only 2% and 5% both have a slight performance drop on supervised evaluation tasks after training for a long time. However, all 4 models keep improving on unsupervised STS14 evaluation task during training until they converge.
> > >
> > > 2/ Larger size of training data requires a longer time to converge, and it generally performs better than those with a smaller size of data.
> > >
> > > It suggests that we need to train our model on Amazon Book Review for more iterations, and our model potentially is able to get even better results. As stated in Radford et al. (2017), their BYTE m-LSTM model was trained on Amazon Review for a month, while we only trained our model for around 33 hours, and we still got comparable results on classification tasks and better results on relatedness and entailment tasks.
> > >
> > > 3/ When it comes to a large model (large dimension of the representation), more data and longer training time will result in better sentence representations.

---

### Author Response · Authors · 2018-01-05
**Revision**

We revised our paper, and it has been updated now. The revision is based on the reviewers’ suggestions.

1/ Additional experiments, suggested by Reviewer2 and Reviewer3, were included to strengthen our original claim, and they are in Section 3 (Architecture Design) now.

2/ We summarized the effect of varying model architecture in Section 3 (Architecture Design), and moved the original Table for quantitative results to the supplementary.

3/ All reviewers recommended to reduce the length of the original Motivation section and Related work section to fit the paper into 8 pages, so we revised these 2 sections to make them concise and precise.

4/ We didn’t change much in the Abstract, the Introduction, and the Conclusion

Overall, the revision didn’t change our main claim, and the additional experiments and compressed paper make our paper even clearer and more concise.

---

### Decision · Program_Chairs · 2018-01-29
**ICLR 2018 Conference Acceptance Decision**

**Decision:**

Reject

**Comment:**

here, yet another sentence representation method is proposed. i agree with R1 and R3 that this does not contribute significantly to be a full-length conference paper.